# Investigating Nutrition and Supportive Care Needs in Esophageal and Gastric Cancer Survivors: A Cross-Sectional Survey

**DOI:** 10.3390/healthcare13162057

**Published:** 2025-08-20

**Authors:** Fatemeh Sadeghi, Juliette Hussey, Suzanne L. Doyle

**Affiliations:** 1Discipline of Physiotherapy, School of Medicine, Trinity College Dublin, D02 PN40 Dublin, Ireland; 2School of Biological, Health and Sports Science, Technological University Dublin, D07 XT95 Dublin, Ireland

**Keywords:** esophageal cancer, gastric cancer, cancer survivorship, oncology nutrition, nutrition care needs, quality of life

## Abstract

**Background/Objectives**: Advances in cancer diagnosis and treatment improved survivorship rates, but survivors’ long-term quality of life remains a critical concern. Survivors of esophageal and gastric cancer often undergo extensive curative surgery, which can have lasting impacts on nutritional status. This study aimed to assess the nutritional status, dietary challenges, and supportive care needs of this population of cancer survivors. **Methods**: In this cross-sectional study, adults diagnosed with esophageal or gastric cancer were invited to complete an anonymous survey to assess their nutritional status, quality of life, and psychological well-being. **Results**: A total of 114 responses were analyzed. Although over 70% of participants were more than two years post-diagnosis, more than 30% remained at risk of malnutrition. Additionally, over 36% reported ongoing dietary complications such as swallowing difficulties, dumping syndrome, diarrhea, and reflux. Impaired quality of life was observed in 31.7% of respondents, defined as having a global health status score below 66.1 on the EORTC QLQ-C30. Psychological distress was also evident, with over 25% screening positive for anxiety or borderline anxiety, and more than 22% for depression or borderline depression. **Conclusions**: Malnutrition, gastrointestinal symptoms, impaired quality of life, and psychological distress remain prevalent among esophageal and gastric cancer survivors many years after diagnosis. These findings underscore the need for ongoing monitoring and the provision of targeted supportive care to improve long-term survivorship outcomes.

## 1. Introduction

Global prevalence of esophageal cancer increased in recent decades, with 0.961 million cases diagnosed in 2019 and 0.54 million related deaths in 2020 [1,2]. Despite declines in gastric cancer incidence in most regions, it remains a public health issue as the fifth most common cancer worldwide [3,4]. Gastric cancer incidences reached >1.2 million cases in 2019, with associated mortality in >950,000 patients [5].

Advances in cancer diagnosis and treatment led to higher survival rates [6]. According to national data at the end of 2021, approximately 215,000 cancer survivors were alive in Ireland, which indicates that 1 in 23 Irish adults is a cancer survivor [7]. The Irish National Cancer Registry reported significant improvements in the 5-year net survival of esophageal and gastric cancer [7]. According to its latest annual report published in 2023, 5-year net survival for esophageal cancer increased to 23% in the years 2014–2018, up from 11% in 1994–1998, while survival for stomach cancer rose to 31% from 17% over the same period [7]. At the end of 2021, over 4000 individuals diagnosed with esophageal cancer and gastric cancer were still alive in Ireland [7].

Despite substantial improvement in survival rate, cancer survivors often do not return to their pre-diagnosis health and suffer from long-term treatment side effects, such as malnutrition, impaired physical function, and psychological problems, which can impair the quality of life in survivorship [8,9,10]. While curative surgery remains the primary treatment for esophageal and gastric cancer, it is accompanied by profound and lasting alterations in gastrointestinal anatomy and function [11]. These changes are associated with long-term persistent side effects such as delayed gastric emptying, early satiety, anorexia, dumping syndrome, reflux, nausea, vomiting, malabsorption, and diarrhea, which can negatively impact their quality of life [11,12]. Furthermore, these nutrition impact symptoms (NIS) limit oral intake and lead to unintentional weight loss and malnutrition, which also impair survivors’ quality of life [13,14,15].

Beyond the physical impairments, the psychological and emotional burden associated with esophageal and gastric cancer is substantial [16,17,18]. Previous studies indicated that patients with esophageal and gastric cancer experience significant psychological distress, beginning at diagnosis and persisting for up to four years after treatment completion [17,19]. Depression and anxiety can lead to noncompliance with treatment and negatively impact survival [20,21]. Furthermore, psychological symptoms are often associated with malnutrition and a reduced quality of life in cancer patients [22,23]. Despite these findings, the psychological impact of cancer diagnosis and treatment remains largely underexplored and insufficiently addressed [16,24,25,26].

To date, there is limited evidence available regarding the unmet nutrition and supportive needs of survivors of esophageal and gastric cancer in Ireland [24]. These data are required to evaluate ongoing challenges faced by survivors beyond the acute treatment phase and to identify areas where additional support or intervention may be warranted. Therefore, the present study aimed to investigate malnutrition risk, gastrointestinal (GI) symptoms, psychological distress, and quality of life in esophageal and gastric cancer survivors. It also explored survivors’ views on unintentional weight loss and the need for ongoing nutrition care and information.

## 2. Materials and Methods

The present study was a cross-sectional observational survey conducted in Ireland between September 2021 and May 2022. Adults aged 18 years and older with a history of esophageal or gastric cancer, who were fluent in English and able to provide informed consent, were included in the study. There was no upper age limit for inclusion to ensure a broad representation of the cancer survivor population. Participants were invited to complete an anonymous survey.

Participants were asked to provide consent by agreeing with two statements of consent at the beginning of the survey questionnaires, which indicated that they had read and understood the participant information leaflet (PIL) and understood that participation is entirely voluntary.

Recruitment took place during the COVID-19 pandemic (September 2021 to May 2022), which posed significant challenges and limited participant enrollment. As a result, multiple strategies were implemented to enhance participation. Initially, participants were invited to complete an online version of the survey. The study was promoted via Irish cancer charity organizations (the Irish Cancer Society and the Oesophageal Cancer Fund) and social media platforms (Twitter, Facebook, and Instagram). A virtual poster and an animation clip were used to introduce and advertise the survey. The advertisement included a link to the Qualtrics website (Qualtrics, Provo, UT, USA), where participants could access the PIL and the survey. By clicking the link, individuals were able to review the study information and, if they chose to participate, complete the survey anonymously on the Qualtrics platform.

Once COVID-19 restrictions eased, eligible patients attending the outpatient upper gastrointestinal (UGI) cancer clinic at St James’s Hospital (SJH) were invited to complete a paper-based version of the survey. Completed surveys could be returned at the clinic or by post using a stamped addressed envelope.

Due to the limited response rate from online and in clinic recruitment, an amendment was submitted to the ethics committee to allow for the direct postal distribution of a paper-based version of the survey to eligible patients.

Subsequently, a postal survey was sent to a random sample of 200 survivors using a GDPR-compliant research mailing list maintained by the upper GI clinic at SJH. A simple random sampling method was employed using a random number generator (random.org, accessed on 30 March 2022) [27]. All individuals on the mailing list previously provided consent to be contacted for research purposes. Prior to mailing the surveys, the list was cross-referenced with death notices published on RIP.ie, an Irish death notice website, to ensure that participants were still living.

The survey consisted of five sections. In Section 1, demographic and clinical data including age, gender, employment status, living arrangements, education level, type of medical insurance, cancer diagnosis, and treatment were collected. In Section 2, nutritional status was assessed using the Malnutrition Screening Tool (MST) [28]. The MST has been reported as a validated tool for predicting malnutrition risk in oncology [29,30]. The MST is an easy and rapid screening tool which includes two questions, examining unintentional weight loss and reduced appetite and nutritional intake. A score of 2 or higher suggests a risk of malnutrition [28]. In Section 2, data on nutritional supplement use, concerns about weight loss, access to dietetic support, perceived need for additional dietetic input, and preferences for nutritional advice were also collected. Relevant literature was reviewed to guide the design of these additional questions [31,32,33,34]. Additionally, an external upper GI dietitian was also consulted in designing questions addressing nutritional issues specific to esophageal and gastric cancer survivors and their care needs during survivorship.

In Section 3, quality of life was measured by the European Organisation for Research and Treatment of Cancer Quality of Life Questionnaire (EORTC-QLQ-C30) [35]. The EORTC QLQ-C30 is a reliable and valid tool designed to assess health-related quality of life in patients with cancer [35]. This tool includes 30 questions and consists of five functional domains (physical, role, cognitive, emotional, and social), eight symptom domains (fatigue, pain, nausea/vomiting, constipation, diarrhoea, insomnia, dyspnoea, and appetite loss), as well as global health/quality-of-life and financial impact [35]. A minimal clinically important difference (MCID) of ≥10 points was considered to determine clinically meaningful differences in EORTC QLQ-C30 scores compared to the general population [36,37]. Additionally, the thresholds for clinical importance (TCI) were considered to identify clinically important symptoms [38].

Section 4 explored GI symptom burden using the Gastrointestinal Symptom Rating Scale (GSRS) [39]. The GSRS is a 15-item questionnaire representing five domains assessing reflux, abdominal pain, indigestion, diarrhoea, and constipation. It asks respondents to recall symptom severity during the last week. It employs a Likert-type scale with seven graded points, where 1 indicates the absence of problematic symptoms, while 7 shows highly problematic symptoms. Consequently, higher GSRS scores indicate more severe and problematic gastrointestinal symptoms. The reliability and validity of the GSRS as a self-administered tool is well documented, and normative values for the general population are available [39,40,41,42,43].

Finally, in Section 5, psychological well-being was assessed using the Hospital Anxiety and Depression Scale (HADS) questionnaire [44]. HADS has been reported to be a valid tool for assessing anxiety disorders and depression and can accurately screen for psychosocial distress in cancer patients [45,46]. HADS is a 14-item questionnaire containing seven questions related to anxiety and seven questions related to depression [46]. A HADS score of ≥11 indicates definite cases of anxiety or depression, scores between 8 and 10 suggest a probable case, and if the score is ≤7, the respondent does not have anxiety or depression.

All questionnaires used in this study were applied in accordance with their respective usage guidelines. Where formal permission was required, it was obtained from the appropriate rights holders. Instruments that are publicly available for academic use were cited accordingly in the manuscript.

The developed questionnaire was then discussed and reviewed by the research team to ensure its alignment with the research objectives.

Statistical analysis was performed using IBM SPSS Statistics for Windows, version 29 (IBM Corp., Armonk, NY, USA). Descriptive analysis was used to present mean and standard deviation (SD) for continuous data and frequency and percentage for categorical data. The Kolmogorov–Smirnov test was used for assessing the normal distribution of data. One sample T-test was used to determine statistically significant differences between sample and reference values. Chi-square and Mann–Whitney U tests were used to compare differences between groups.

## 3. Results

The recruitment process and response rate are presented in Figure 1. Due to the nature of online recruitment in the survey, it was not possible to determine the exact number of individuals who viewed the advertisement and met the eligibility criteria. Despite frequent re-advertising over a three-month period, only five responses were received through the online survey and included in the analysis. In-clinic recruitment took place over six weeks, with one clinic scheduled per week. While the exact number of patients approached in clinic during this period was not recorded, this strategy yielded eight complete surveys; however, two responses were excluded from analysis as participants had no confirmed diagnosis of esophageal or gastric cancer. Of the 200 posted survey packets, 106 patients (53%) returned the survey. From this, three responses were excluded from the analysis, as one participant returned a questionnaire belonging to another research study and two returned blank questionnaires.

Overall, one hundred and fourteen (114) survey responses from three recruitment pathways were included in the analysis.

Participants’ characteristics are presented in Table 1. The majority of participants were male (n = 72, 63%). The mean age was 70.36 years ± 9.19 (41–88 years), and the majority were aged over 65 years (n = 82, 71.9%). Three quarters (75.4%) were esophageal cancer survivors, and more than 70% of participants were >2 years post-diagnosis. Almost all participants underwent curative surgery (99%), and 42% were no longer receiving cancer treatments at the time of the study.

As presented in Table 2, one-third of the participants were identified as being at risk of malnutrition. Subgroup analysis showed that the mean MST scores and the proportion of participants at risk of malnutrition were higher among males and gastric cancer survivors; however, these differences were not statistically significant. There was no statistically significant difference in mean MST score and risk of malnutrition between different sociodemographic factors. However, higher MST scores and a greater proportion of participants at risk of malnutrition were noted among those living alone (42.9% vs. 27.9%), those not in employment (31.4% vs. 16.2%), and those with public health insurance (38.3% vs. 24.5%) compared to their respective counterparts.

Over 36% of participants reported ongoing nutrition impact symptoms and complications. Reflux, diarrhea, dumping syndrome, and swallowing difficulty were the most reported GI complications.

As presented in Table 3, study subjects scored significantly higher on all five domains of GSRS compared to the general population, indicating the higher prevalence and severity of gastrointestinal complications in survivors. Additionally, diarrhea syndrome and the total GSRS score exceeded the minimal clinically important difference (MCID) [47]. Subgroup analysis of GSRS scores showed a statistically significant difference in reflux syndrome scores, with females reporting a higher mean score (2.34) compared to males (1.83). Despite being statistically significant, this difference did not reach the threshold of MCID [47]. No significant differences were observed between esophageal and gastric cancer survivors across any GSRS subscales.

Table 4 summarizes the EORTC QLQ-C30 quality of life data. Compared to the general population and considering minimal clinically important difference (MCID) of ≥10 points [36,37], respondents reported impaired physical and social functioning.

Additionally, a high symptom burden was observed, particularly for nausea and vomiting, dyspnea, appetite loss, constipation, diarrhea, and financial difficulties. Notably, 31.7% of respondents had a global health status score below 66.1, indicating impaired quality of life [48,49]. Additionally, considering thresholds for clinical importance (TCI) of EORTC scores, physical function, nausea and vomiting, dyspnea, and diarrhea may require clinical attention in this cohort of UGI cancer survivors [38].

Subgroup analysis of EORTC scores by gender showed that women reported statistically significantly lower physical functioning and higher symptom scores for appetite loss and constipation compared to men. However, only the differences in appetite loss and constipation reached the MCID of ≥10. EORTC scores did not differ significantly between esophageal and gastric cancer survivors, except for insomnia, which was significantly higher among gastric cancer survivors (41.33 ± 37.61 vs. 24.31 ± 28.81, *p* = 0.038).

**Table 4 healthcare-13-02057-t004:** Participants’ quality of life (EORTC QLQ-C30).

	Scale	Sample Mean (SD)	Reference Values [50] General Population
Global Health Status/QoL	QoL (n = 107)	66.27 (24.21)	71.2 (22.4)
Functional scales ^1^	Physical functioning (n = 111)	79.50 (20.80)	89.8 (16.2)
	Role functioning (n = 112)	77.67 (28.61)	84.7 (25.4)
	Emotional functioning (n = 111)	78.3 (26.61)	76.3 (22.8)
	Cognitive functioning (n = 111)	79.27 (20.25)	86.1 (20)
	Social functioning (n = 111)	74.62 (26.57)	87.5 (22.9)
Symptom scales ^2^	Fatigue (n = 111)	31.83 (21.25)	24.1 (24)
	Nausea and vomiting (n = 112)	13.24 (20.88)	3.7 (11.7)
	Pain (n = 111)	15.76 (22.90)	20.9 (27.6)
	Dyspnea (n = 112)	19.94 (27.01)	11.8 (22.8)
	Insomnia (n = 111)	27.92 (31.63)	21.8 (29.7)
	Appetite loss (n = 111)	22.52 (32.46)	6.7 (18.3)
	Constipation (n = 112)	14.88 (24.02)	6.7 (18.4)
	Diarrhea (n = 110)	23.63 (29.03)	7.0 (18)
	Financial difficulties (n = 110)	17.57 (26.60)	9.5 (23.3)

QoL, quality of life. SD, standard deviation. EORTC, European Organisation for Research and Treatment of Cancer. ^1^ The score ranges from 0 to 100; a higher score represents a better quality of life or a higher level of functioning. ^2^ The score ranges from 0 to 100, and a higher score represents more severe symptoms. Sample size varies across scales due to missing responses. In the event of missing data, the developer’s recommended guidelines for handling and imputing missing values were followed [35].

Table 5 presents the results of the HADS questionnaire. The study respondents had lower anxiety scores compared to the normative value for the general population but scored higher on the depression scale. As shown in Figure 2, over 25% were identified with anxiety or borderline anxiety, and over 22% of respondents had depression scale scores indicating depression or borderline depression.

Subgroup analysis of HADS scores showed that females had significantly higher anxiety and total scores compared to males. Additionally, survivors of gastric cancer had a statistically significantly higher depression score compared to esophageal cancer survivors.

Collected data on respondents’ nutrition care needs are presented in Table 6. Almost one-third (29.8%) of respondents reported experiencing weight loss, and among them, over 75% expressed concern about it. Overall, more than half of all respondents reported some level of concern regarding weight loss. Although approximately 60% of respondents sought dietetic advice, >50% agreed that they would benefit from further contact with a dietitian. Similarly, nearly 60% of respondents expressed interest in additional information on nutrition. The most reported subjects of interest for nutritional information were symptom management, recipes and meal plans, supplements, healthy eating, and portion sizes.

## 4. Discussion

The present study explored the risk of malnutrition, severity of GI symptoms, quality of life, and psychological well-being in esophageal and gastric cancer survivors. It also investigated survivors’ perceptions of unintentional weight loss, ongoing nutrition care, and information needs. The findings indicate that malnutrition and GI symptoms remain prevalent among this cohort of cancer survivors. The results of the present study show impaired quality of life and the presence of psychological distress among survivors. Additionally, the majority of esophageal and gastric cancer survivors had a positive attitude towards further dietary support and a strong interest in receiving additional nutritional information.

### 4.1. Malnutrition

In the present study, one in three participants were found to be at risk of malnutrition. Evidence shows patients undergoing surgery for gastrointestinal cancer are at higher risk of malnutrition compared to other cancer patients [52,53]. The prevalence of malnutrition in the present study is comparable to the recent national data reporting risk of malnutrition in 33.8% of hospital patients, highlighting the considerable and ongoing nutritional vulnerability of esophageal and gastric cancer survivors [54]. The prevalence of malnutrition in GI cancers ranges from 22% to 85%, depending on factors such as the stage of cancer, cancer treatment, and survivorship phase [52,55,56,57]. Some studies suggested a lower malnutrition risk and reduction in the pace of weight loss at 12+ months post-surgery for esophageal and gastric cancer [58,59]. This is notable, as the results in the present study may be influenced by the fact that over 73% of respondents were survivors of more than two years.

Although unintentional weight loss (UWL) is frequently reported in esophageal and gastric cancers, information on patient’s subjective experience and perception of weight loss is limited [60]. In the current study, almost one-third (29.8%) of respondents reported weight loss. A national survey by O’Sullivan et al. reported a higher prevalence of weight loss in a mixed cohort of cancer survivors, with 44% of participants experiencing weight loss [31]. However, in the present study, over 75% of respondents who experienced weight loss expressed concern about it, compared to 42% of participants in the study by O’Sullivan et al. The lower rate of concerns about UWL in the O’Sullivan study may be attributed to the recruitment of a mixed cohort of cancer survivors, with over 50% being breast cancer survivors. For breast cancer survivors, being overweight or obese is often a risk factor for overall survival, which may reduce the level of concern about weight loss [61]. In contrast, for patients with esophageal and gastric cancers, overweight or obesity can be a protective factor for overall survival, and UWL is a significant risk factor for poor overall survival [61]. Additionally, as noted earlier, UWL is highly prevalent in esophageal and gastric cancer; hence, it remains a key concern for many patients, their caregivers, and the healthcare team throughout their cancer treatment [13,62,63]. The higher rate of concern about weight loss in the present study may be a reflection of the acute awareness that patients have regarding weight loss and its implications. It may suggest that there is a higher psychological burden of UWL in this cohort of cancer survivors.

The findings of the present study, aligned with other best practice guidance, highlight that nutrition care services should ensure routine malnutrition risk screening and comprehensive assessment to deliver timely personalized nutrition care in esophageal and gastric cancer [64].

In the present study, descriptive trends suggested that participants living alone, those not in employment, and those who had public health insurance had higher MST scores and a greater proportion of them were at risk of malnutrition. While these differences did not reach statistical significance, they may indicate underlying socio-economic disparities in nutritional vulnerability among cancer survivors. Data on demographic and social determinants of malnutrition in esophageal and gastric cancer survivors are scarce and inconsistent. Some studies found that low socioeconomic status (SES) may be linked to a higher risk of malnutrition in cancer patients [65,66,67,68]. However, data on SES were not collected in the present study due to the complexity of SES indicators and challenges with using household income to examine SES [69]. Future research is required to investigate the impact of SES on malnutrition risk in cancer survivors in Ireland. Such data will be valuable in designing targeted interventions for those at the highest risk of malnutrition.

### 4.2. Nutrition Care and Information Needs

The present study also obtained respondents’ perceptions of nutrition care and information needs. The findings indicate that approximately 60% of respondents sought advice from a dietitian at some stage of their cancer treatment. Based on a national survey conducted by O’Sullivan, patients with GI cancers were most likely to see a dietitian during their cancer journey compared to other cancer types (71.8%) [31]. In the present study, over 50% of respondents had a positive view of further dietetic input and expressed interest in receiving additional nutritional information. These findings are consistent with previous studies, highlighting the unmet need for nutrition information among cancer survivors [31,70,71,72,73]. Despite understanding the important role of nutrition in cancer recovery and their strong interest in receiving nutritional information, many still lack access to the information they need [31,70,71,72]. However, accessing reliable, evidence-based information can be challenging due to the current staffing shortages in dietetic services across Ireland [74]. The application of e-health in delivering nutrition care is suggested to be clinically beneficial and have the potential to overcome several barriers to delivering nutrition care, such as time and resource restraints [75]. Digital resources such as web-based information can empower patients to make dietary choices and manage their cancer-related symptoms [76,77].

### 4.3. Gastrointestinal Symptoms

The results of the present study indicate the persistence of GI symptoms in esophageal and gastric cancer survivors, as respondents scored higher in all domains of GSRS compared to the general population. A similar study reported long-term persistence of GI disturbance, including early satiety, regurgitation, and heartburn more than five years after esophagectomy [78]. Reflux, diarrhea, dumping syndrome, and swallowing difficulty were the most commonly reported symptoms in the current study, and similar findings have been observed in other studies [58,79]. Aligned with previous studies, the findings of the present study further support the need for tailored interventions to address the burden of GI symptoms in this cohort of cancer survivors. It is a recommended to integrate patient-reported outcomes (PROs) into survivorship care to tailor the care plan to survivors’ individual needs. Incorporating PRO measures can facilitate personalized care and reduce the burden on healthcare professionals and should be incorporated into clinical practice as a standard [80]. Several studies have shown that PROs more accurately capture patient symptoms compared to physician’s assessment [81].

### 4.4. Quality of Life

The adverse impact of esophageal and gastric cancers on health-related quality of life (HRQOL) is well documented [82,83,84]. Long-lasting and possible lifelong impairment of quality of life and persistence of symptoms following esophageal and gastric cancer have been observed in previous studies [85,86,87,88,89]. The findings of the present study further indicate the persistence of GI symptoms post-esophageal and gastric cancer, which may affect HRQOL. The results of the present study demonstrate impaired quality of life in 31.7% of respondents, comparable to a previous study, which reported impaired quality of life in 26% of esophageal cancer survivors between 5 and 10 years post-operatively [86]. Interestingly, an Irish study by Bennett et al. reported impaired quality of life in 20% and 14.3% of the patients attending a nutrition survivorship clinic at 6 and 12 months post-esophagectomy, respectively [48]. This may highlight the benefits of nutrition care in symptom management and improving the quality of life of cancer survivors.

Incorporating regular monitoring of HRQOL into standard care can provide valuable insights into how cancer and its treatment affect patients’ overall well-being, helping to track treatment toxicity and effectiveness and assisting clinicians in delivering personalized care tailored to each patient’s needs and preferences [90]. Incorporating PROs with predefined thresholds for intervention in clinical practice leads to significant improvements in symptom management and quality of life [91,92,93,94]. However, further research is warranted to examine current practices, identify potential challenges, and the necessary infrastructure required for integrating HRQOL monitoring into routine clinical practice, ensuring it becomes a practical and effective part of patient care.

### 4.5. Psychological Status

Higher rates of anxiety are reported in cancer survivors compared to the general population [95]. The prevalence of anxiety and depression in patients with cancer varies in different studies and ranges from <10% to 30% [96]. Several factors, such as type and stage of cancer and cancer treatment, may impact patients’ psychological status and explain the variability in the prevalence of anxiety and depression reported in studies [97,98].

In the present study, 25.9% and 22.3% of patients scored above the cut-off for a possible–probable anxiety and depression disorder, respectively. These findings were comparable to a previous study, which observed that 24% of patients reported depression at a median of 45 months following diagnosis of esophageal cancer [19]. A longitudinal study including both esophageal and gastroesophageal junction cancer patients reported anxiety and depression rates of 24% and 27%, respectively, at 12 months post-diagnosis [17]. Another longitudinal study involving patients with esophageal and gastroesophageal junction cancer found anxiety rates of 33% before surgery, 28% at six months post-operatively, and 37% at 12 months post-operatively. It also reported that depression rates rose from 20% pre-surgery to 27% at six months and 32% at 12 months post-operatively [99]. Similarly, a recent study highlighted a rise in anxiety among esophago-gastric cancer survivors between 3 and 6 months post-discharge, with levels exceeding those reported at the time of diagnosis [100].

While available evidence on psychological morbidity in esophageal and gastric cancer survivors provide valuable context, it is important to note that cultural factors, differences in access to mental health services, and social support may impact rates of psychological complications reported in studies from different countries [101,102]. Thus, it is important to consider the psychological morbidity in the general Irish population when interpreting the results of the present study. In the present study, 11.6% of respondents had a HADS-anxiety score of ≥11 and 7.1% had a HADS-depression score of ≥11, indicating moderate to severe depression and anxiety. According to the Mental Health Association of Ireland (2001), the prevalence of moderate to severe anxiety and depression in the Irish population using HADS was 13% and 4%, respectively [103]. Although the rate of anxiety in the Mental Health Association report was similar to the findings of the present study, compared to recent data from The Irish Longitudinal Study on Ageing (TILDA), a higher score of anxiety was observed in the present study, considering a minimal clinically important difference (MCID) of two points (5.26 ± 4.48 vs. 3.14 ± 312) [104,105]. Moreover, in the present study, 22.3% of cancer survivors were categorized as borderline or clinically depressed compared to 11% of those aged 65 years and over in an Irish study by McGee et al. [106]. Of note, in the present study, six (5.35%) participants had both HADS-D and HADS-A ≥ 11, compared to 3.4% of participants in an Irish study of a mixed cohort of oncology patients [107]. The findings of the present study demonstrate higher psychological morbidity in esophageal and gastric cancer survivors compared to the general population, which highlights the importance of incorporating targeted psychological support into survivorship care plans.

Psychological complications can limit treatment adherence, which not only impacts patients’ survival, but also has the potential to dilute data on the effectiveness of interventions and underestimate the efficacy of therapies [21]. Symptoms of depression and anxiety have been linked to poor adherence to dietary recommendations [108]. Additionally, depression may impair patients’ capacity to understand and process information regarding prognoses [20]. Incorporating psychological assessment into standard cancer care and survivorship may provide more accurate information on the prevalence of psychological morbidities and psychological consequences of cancer [21]. It also may facilitate access to required services for those affected, which subsequently may improve treatment adherence, quality of life, and survival [21].

### 4.6. Strengths and Limitations

To the best of our knowledge, this cross-sectional survey is the first study in Ireland investigating esophageal and gastric cancer survivors’ nutrition and supportive care needs. In the present study, the respondents’ demographic was reflective of the age and gender profile of esophageal and gastric cancer survivors in Ireland, which may improve the reliability of the findings in representing survivorship and nutrition care needs in this cohort of cancer survivors. In the present study, 72% of survivors were over 65 years old, which is comparable to the age profile at diagnosis according to national data. Additionally, the majority of participants were male, which is reflective of the higher incidence of these cancer types in men, as over 60% of new cases are male [7]. Moreover, most participants were more than two years post-diagnosis, enabling an exploration of supportive care needs in long-term survivorship. Using validated tools to examine the risk of malnutrition (MST), GI symptoms (GSRS), QOL (EORTC-30), and psychological status (HADS) is another strength of this study.

However, the results of the present study should be interpreted with caution, considering the following limitations. The findings reflect data from patients who responded to the survey; thus, the possibility of non-response bias may limit the generalization of findings to the broader population of esophageal and gastric cancer survivors. There may also be an element of responder bias in that those with a particular concern regarding their nutritional status or ongoing symptoms were more likely to complete the survey. We also acknowledge that the findings of this study may have limited generalizability due to potential variations in healthcare systems, nutritional support services, and cancer care pathways across different countries.

Recruitment for the present study took place during the COVID-19 pandemic. Although services were beginning to gradually normalize during study period, healthcare disruptions during COVID-19 pandemic may have affected participants’ access to care, potentially influencing self-reported outcomes such as malnutrition risk, gastrointestinal symptoms, quality of life, and psychological well-being. It is important to note that majority of responses were received during April–May 2022, when restrictions had largely been lifted, which may have reduced the variability of the findings. However, the broader COVID-19 context may still have impacted the findings, and this should be considered when interpreting the findings.

It should be noted that the online version of this survey struggled to recruit participants despite repeated advertisements through various social media platforms. This may be due to the demographic profile of esophageal and gastric cancer survivors, as the majority of diagnosed cases are >65 years of age, and they may have lower digital literacy [109]. Additionally, it was reported that in 2019, 33% of the Irish population aged between 65 and 74 never accessed the internet, and nearly half of individuals aged 64–75 never used the internet [110,111]. These factors should be considered while designing future studies in this cohort of cancer survivors.

## 5. Conclusions

Results of the present study indicate a significant risk of malnutrition, persistent GI complications, impaired quality of life, and psychological burden in esophageal and gastric cancer survivors. The findings highlight ongoing challenges in the long-term recovery following these cancers and the need for continued monitoring of outcomes and provision of supportive care for survivors.

Survivors of esophageal and gastric cancer experience complex, multifaceted needs across the cancer continuum, which warrant further research to identify stage-specific challenges and tailor interventions accordingly. Multidisciplinary care programs integrating nutritional, psychological, and symptom management are recommended to address these diverse survivorship needs. However, the optimal structure and effectiveness of such programs require further investigation.

To improve outcomes in this population, it is imperative that patient-focused, multidisciplinary interventions be designed, implemented, and rigorously evaluated. This study provided preliminary data that may inform the development of nutrition and supportive care components within these programs and supports the broader call for integrated survivorship care in gastrointestinal oncology.

## Figures and Tables

**Figure 1 healthcare-13-02057-f001:**
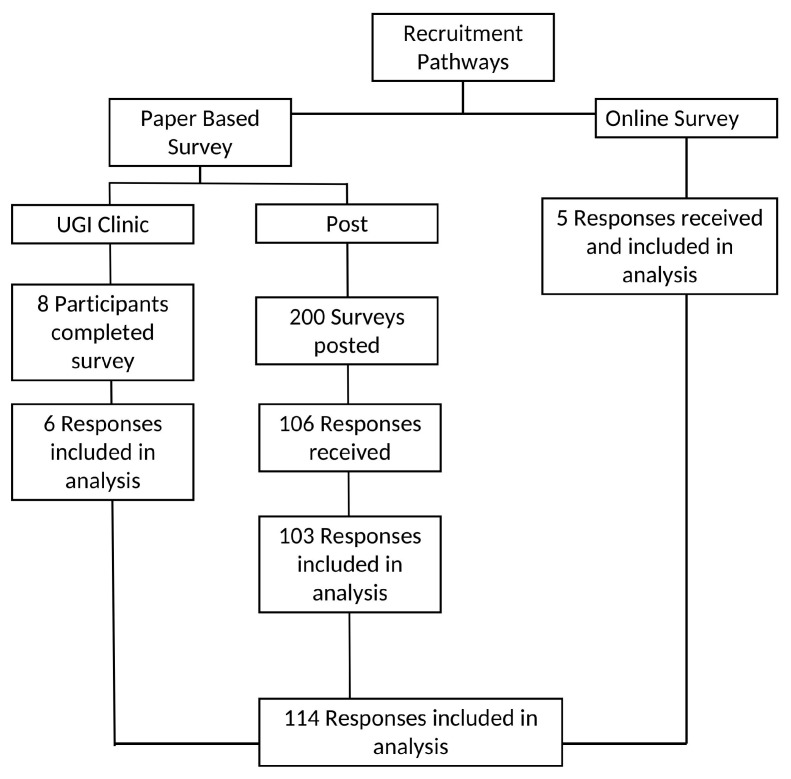
Recruitment pathways and response inclusion process.

**Figure 2 healthcare-13-02057-f002:**
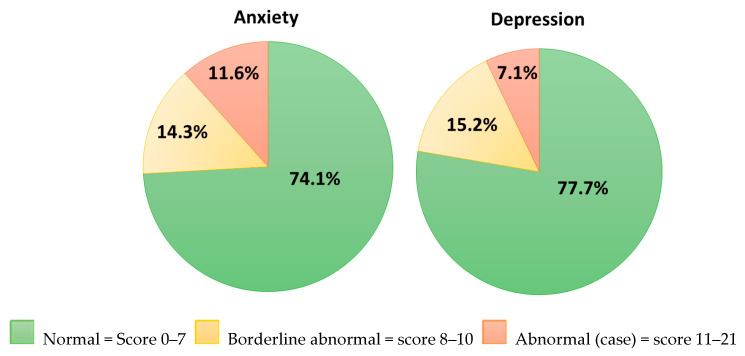
Hospital anxiety depression scale score.

**Table 1 healthcare-13-02057-t001:** Participants’ characteristics.

		n (%)
Age (years) mean ± SD, range, (n = 110)	70.36 ± 9.19, 41–88	-
Gender, (n = 114)	Female	42 (36.8)
	Male	72 (63.1)
Living arrangement, (n = 114)	Living with others	93 (81.6)
	Living alone	21 (18.4)
Employment status, (n = 114)	Employed	24 (21)
	Not employed	90 (78.9)
Ethnicity, (n = 114)	Irish	107 (93.8)
	Other White backgrounds	7 (6.1)
Health insurance, (n = 114)	Public	47 (41.2)
	Private	61 (53.5)
	None	6 (5.2)
Education, (n = 112)	University degree	39 (34.8)
	No university degree	73 (65.17)
Cancer diagnosis	Esophageal cancer	86 (75.4)
	Gastric cancer	28 (24.5)
Time from diagnosis (months) mean ± SD, (n = 108)	44.76 ± 34.18	-
	Range: 2–270	
Cancer treatment, (n = 114)	Surgery only	43 (37.7)
	Chemotherapy only	1 (0.87)
	Surgery + chemotherapy/radiotherapy/immunotherapy	70 (61.4)
	No longer receiving treatment	48 (42.1)
	On a break from cancer treatment	4 (3.5)

Sample size varies across variables due to missing responses.

**Table 2 healthcare-13-02057-t002:** Participant’s nutritional status screened by the malnutrition screening tool (MST), n = 114.

Nutritional Status		n (%)
MST score (mean ± SD)	1.11 ± 1.51	-
Risk of malnutrition	Low	79 (69.29)
	Medium	21 (18.42)
	High	14 (12.28)
Ongoing nutrition impact symptoms	Yes	42 (36.84)
	No	72 (63.15)

**Table 3 healthcare-13-02057-t003:** Gastrointestinal symptoms rating scale score.

GSRS Domains Score	Sample			Reference Value, General Population [43]	*p*-Value
	n	Mean Score (SD)	CI	Mean Score	
Abdominal pain syndrome (score range 1–7)	109	1.85 (1.01)	1.65, 2.04	1.56	0.004
Reflux syndrome (score range 1–7)	108	2.05 (1.41)	1.78, 2.32	1.39	<0.001
Diarrhea syndrome (score range 1–7)	113	2.40 (1.59)	2.11, 2.70	1.38	<0.001
Indigestion syndrome (score range 1–7)	111	2.15 (1.24)	1.91, 2.38	1.78	0.002
Constipation syndrome (score range 1–7)	113	1.93 (1.23)	1.70, 2.16	1.55	0.001
Total score	108	2.06 (1.02)	1.86, 2.25	1.53	<0.001

Sample size varies across domains due to missing responses. The mean value for the items in each dimension was calculated to score the GSRS. In the event of missing data, the developer’s recommended guidelines for handling and imputing missing values were followed.

**Table 5 healthcare-13-02057-t005:** Hospital anxiety depression scale score.

	Sample Mean (SD) (n = 112)	Normative Value (General Population) Mean (SD) [51]
Anxiety scale	5.26 (4.48)	6.14 (3.76)
Depression scale	4.72 (3.81)	3.68 (3.07)
Total score	9.98 (7.70)	9.82 (5.98)

**Table 6 healthcare-13-02057-t006:** Participants reported nutrition care needs.

Nutrition Care Needs	n (%)
Concern about weight loss (n = 110)	
Not concerned at all	51 (46.3)
A bit concerned	31 (28.18)
Moderately concerned	18 (16.36)
Severely concerned	10 (9.09)
Sought advice from a dietitian (n = 109)	
Yes	63 (57.7)
No	46 (42.2)
Would benefit from further dietitian contact (n = 111)	
Yes	58 (52.25)
No	53 (47.74)
Expression of interest in nutrition information (n = 111)	
Interested	64 (57.65)
Not interested	47 (42.34)

## Data Availability

The original contributions presented in this study are included in the article. Further inquiries can be directed to the corresponding author.

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
