# Peer review of "Investigating Nutrition and Supportive Care Needs in Esophageal and Gastric Cancer Survivors: A Cross-Sectional Survey"

_healthcare, 2025, doi:10.3390/healthcare13162057_

Round 1

Reviewer 1 Report

Comments and Suggestions for Authors

Lines 37–41 mention cancer survivors, but no specific values or statistics are provided for survivors of the particular cancer type under study. It would be better to specify this information more clearly.

In Materials and Methods, participants are mentioned as "18 and older," but it is not clear up to what age. Please clarify the upper age limit.

In line 70, the methodology states that patients first made contact via social media and then in person. It is important to specify how many participants contacted via social media versus in person. Also, clarify whether the identity of the social media profiles was verified and if verbal or written informed consent was obtained.

It is necessary to mention whether the study was approved by an ethics committee. Even if exploratory, this ensures data anonymity and ethical compliance.

In Figure 1, some methodological points are addressed, but several adjustments are needed:

  • How was it verified that the same patient did not respond multiple times using different profiles?
  • The image quality is somewhat low.
  • The diagram states there were 106 responses, then later refers to 103—why were 3 responses excluded? Please specify the inclusion/exclusion criteria.

How long was the response collection period, and could this period have influenced the results?

The sample size (n) varies across different patient characteristics (e.g., 112, 114, etc.). The table should be clarified or detailed enough to explain these differences.

Were subgroup analyses by gender performed for anxiety and depression? Similarly, for quality of life?

Additionally, if most participants were older adults, was information gathered on whether they lived alone, with family, or in assisted living? This could impact their nutritional status. Also, was it assessed whether they prepared their own food or if someone else did it for them?

How many patients showed anxiety and depression according to the HADS cut-off points (8 and 7, respectively)? This would help to better interpret the findings. Furthermore, would it be possible to calculate the relative risk (RR) or odds ratio (OR) for malnutrition associated with anxiety or depression?

Was there any subgroup analysis for esophageal versus gastric cancer patients, or were they analyzed as a single group?

Did you evaluate other variables such as erotophilia or erotophobia, given their possible link with anxiety and/or depression? Also, would it be possible to classify patients by socioeconomic level, as this might be related to their nutritional status?

The conclusion mentions a high risk, but the results section does not show any actual risk calculation. Please either calculate the risk or reformulate the conclusion accordingly.

The references do not appear to follow the journal’s author guidelines. Please revise them accordingly or use suitable reference management software.

Several results could be better presented using graphs or visual aids, which would improve the readability of the paper. The topic is highly interesting, especially because this patient group is rarely studied—most research focuses on cancers such as breast cancer. This work could help others interpret their own findings in this underexplored population, whose data is indeed very relevant.

Reviewer 2 Report

Comments and Suggestions for Authors

I appreciate the opportunity to evaluate this article and compliment the authors for their work. While the study presents relevant findings, my main concern relates to the methodological aspects. In addition, I have provided further suggestions.

Title

  • The current title, “Evaluating Nutrition and Supportive Care Needs in Esophageal and Gastric Cancer Survivors,” does not clearly states the nature or design of the study. I recommend that the authors consider rephrasing the title.

Abstract

  • In the abstract, the Methods section does not specify the type of study, which may lead to confusion. Additionally, the phrase “global health status score below 66.1” is unclear, particularly for those unfamiliar with this metric. I recommend either rephrase or provide a brief explanation.
  • Line 25 – the word “many” is in bold.

Introduction

  • Lines 46 and 57 – Please ensure there is a space between the reference in brackets and the preceding word.
  • The Introduction offers a relevant contextualization of the topic and addresses its key aspects. It includes statistics concerning this patient population, primarily in the context of Ireland, where the study was conducted. However, for the benefit of international readers, it would be helpful to include some information or context on the global prevalence and impact of esophageal and gastric cancers.

Materials and Methods

  • As with the title and abstract, the Materials and Methods section does not clearly state the type of study conducted. In fact, the section begins directly with the inclusion criteria. Before addressing this, other essential elements, such as the study design, setting, and timeframe, should be presented to help orient the reader. I recommend that the authors consider following established reporting guidelines, such as the STROBE (Strengthening the Reporting of Observational Studies in Epidemiology) statement, to improve the clarity and structure of their reporting. Several key methodological details are currently missing or insufficiently described, for example, the overall population size and the specific dates or timeframe of data collection. It appears that data may have been collected during and after the COVID-19 pandemic, which could have influenced participants' experiences and responses. While the instruments used in the study are listed, additional information is needed regarding their use. Specifically, it should be stated whether permission was obtained from the original authors to use the instruments, whether they are publicly available, and whether their psychometric properties (e.g., validity, reliability) were evaluated or confirmed in the context of this study population. The absence of this information represents a significant limitation in the reporting of the study.

Results

  • The authors state that 119 responses were considered; however, Figure 1 indicates that only 114 responses were included in the final analysis. This discrepancy should be clarified.
  • Figure 1 is not labelled appropriately, as it illustrates not only the recruitment process but also the exclusion of responses based on predefined criteria (e.g., confirmed diagnosis of esophageal or gastric cancer). I suggest renaming or relabeling the figure.
  • Table 1 presents participant characteristics, including education level. However, the way educational categories are reported may not be easily understood by international readers, as education systems vary widely across countries. I recommend that the authors clarify or rephrase these categories, either by providing a brief explanation or aligning them with internationally recognized educational levels.
  • Lines 158–161: The authors state, “More than half of respondents were concerned about weight loss to some extent. Almost one-third (29.8%) of respondents reported weight loss, of which >75% reported that they were concerned about weight loss.” The second sentence partially repeats the idea presented in the first. I suggest rephrasing this to avoid redundancy.
  • The results are generally well presented and organized according to the type of data and instrument used. However, it would be beneficial to consider cross analyzing the findings from the different instruments to provide a more comprehensive and integrated understanding of the study outcomes. Alternatively, please clarify if such an analysis is planned for a subsequent publication.

Discussion

  • Line 184 - Please ensure there is a space between the reference in brackets and the preceding word.
  • The Discussion section is well aligned with the results and thoughtfully compares the main findings with those of other studies in the field. However, I felt it would benefit from a more detailed discussion on practical implications. Specifically, based on your results, what recommendations can be made for healthcare professionals who support these patients?

Conclusion

  • The conclusions presented are quite brief and general. Expanding this section would strengthen the manuscript by more clearly articulating the key findings, their implications for health professionals, and how they contribute to the existing literature.

References

  • The references cited are generally appropriate and relevant. However, I suggest aiming for a more balanced distribution in terms of publication dates. Currently, approximately 40% of the references are over 10 years old, while only 28% were published within the last 5 years.
  • Reference 49 is missing a publication date. Please revise

Round 2

Reviewer 1 Report

Comments and Suggestions for Authors

The authors have satisfactorily addressed all of my comments, which contributes to improving the quality of the manuscript.

Reviewer 2 Report

Comments and Suggestions for Authors

I have read the revised manuscript. The authors have thoroughly addressed all of my comments. The overall quality and clarity of the manuscript have been significantly improved. I consider it appropriate for publication in its present version.